

**Anomalous acceleration of mass loss in the Greenland ice sheet**
**drainage basins and its contribution to the sea level fingerprints**
**during 2010－2012**
**Linsong Wang[1,2], Liangjing Zhang[2], Chao Chen[1], Maik Thomas[2,3], and Mikhail K. Kaban[2]**
[1] Hubei Subsurface Multi-scale Imaging Key Laboratory, Institute of Geophysics and Geomatics, China
University of Geosciences, Wuhan 430074, China
[2] Helmholtz Centre Potsdam, GFZ German Research Centre for Geosciences, Telegrafenberg, Potsdam
14473, Germany
[3] Institute of Meteorology, Freie Universität Berlin, Berlin, Germany
**Correspondence:** Linsong Wang (wanglinsong@cug.edu.cn)
**Abstract**. The sea level rise contributed from ice sheet melting has been accelerating
due to global warming. Continuous melting of the Greenland ice sheet (GrIS) is a
major contributor to sea level rise, which impacts directly on the surface mass balance
and the instantaneous elastic response of the solid Earth. To study the sea level
fingerprints (SLF) caused by the anomalous acceleration of the mass loss in GrIS can
help us to understand drivers of sea level changes due to global warming and the
frequently abnormal climate events. In this study, we focus on the anomalous
acceleration of the mass loss in GrIS at the drainage basins from 2010 to 2012 and on
its contributions to SLF and relative sea level (RSL) changes based on self-attraction
and loading effects. Using GRACE monthly gravity fields and surface mass balance
(SMB) data spanning 13 years between 2003 and 2015, the spatial and temporal
distribution of the ice sheet balance in Greenland is estimated by mascons fitting
based on six extended drainage basins and matrix scaling factors. Then the SLF
spatial variations are computed by solving the sea level equation. Our results indicate
that the total ice sheet mass loss is contributed from few regions only in Greenland,



i.e., from the northwest, central west, southwestern and southeastern parts. Especially
along the north-west coast and the south-east coast, ice was melting significantly
during 2010–2012. The total mass loss rates during 2003–2015 are −288±7 Gt/yr
and −275±1 Gt/yr as derived from scaled GRACE data and SMB respectively; and
the magnitude of the trend increased to −456±30 Gt/yr and to −464±38 Gt/yr
correspondingly over the period 2010–2012. The residuals obtained by GRACE after
removing SMB show a good agreement with the surface elevation change rates
derived from pervious ICESat results, which reflect a contribution from glacial
dynamics to the total ice mass changes. Melting of GrIS results in decreased RSL in
Scandinavia and North Europe, up to about −0.6 cm/yr. The far-field peak increase
is less dependent on the precise pattern of self-attraction and loading; and the average
global RSL was raised by 0.07 cm/yr only. Greenland contributes about 31% of the
total terrestrial water storage transferring to the sea level rise from 2003 to 2015. We
also found that variations of the GrIS contribution to sea level have an opposite V
shape (i.e., from rising to falling) during 2010–2012, while a clear global mean sea
level drop also took place (i.e., from falling to rising).

**Key words**. GRACE; SMB; Greenland ice sheet; anomaly melting; sea level
fingerprints



## 1 Introduction


The sea level rise due to melting of ice sheets, glaciers and ice caps has been
accelerating in consequence of global warming. The mass change of polar ice sheets
is a major global concern, especially due to its direct impact to global sea level rise
(Forsberg et al., 2017). Estimation of the global ice balance has been obviously
improved in recent years based on available satellite observations, model simulations
and the development of data processing technologies, e.g., using the Gravity
Recovery and Climate Experiment (GRACE) (Rodell et al., 2009; Jacob et al., 2012;
Velicogna et al., 2014) and the Ice, Cloud, and land Elevation Satellite (ICESat)
(Zwally et al., 2011; Shepherd et al., 2012; Gardner et al., 2013). In the last decade,
most studies have confirmed that significant mass loss takes place in the ice sheets of
Greenland and Antarctica, which corresponds to approximately 7 m and 57 m of the
sea level rise respectively when the mass is completely melted (Bamber et al., 2001;
Lythe et al., 2001). Therefore, there is a high demand to monitor the trend in mass
balance changes over Greenland and Antarctica to better understand global climate
change and associated sea level rise.
Due to global warming, frequency and intensity of extreme weather events (i.e.,
snowstorms, cold currents, torrential rains, heat waves, etc.) are increasing globally.
Since the early 1990s, satellite data show that the global mean sea level has been
rising by about 3 mm/yr. Numerous scientific papers on ice sheet changes and their
contribution to sea level rise have been published based on satellite observations over
the last decade, but we still need to focus on the continental ice mass balance caused
by abnormal climate fluctuations in a short term period. A solitary wave disturbance
of global mean sea level has happened during 2010–2012, when the sea level
decreased by 5 mm from the beginning of 2010 to mid 2011 and then rose by nearly
20 mm until the end of 2012 (NASA: SEA LEVEL CHANGE Observations from
Space). This occurred along with a La Niña phase of the El Niño–Southern
Oscillation (ENSO). Previous studies have shown that the change in the sea level





during La Niña is related to water temporarily moved from the oceans to the land,
when precipitation increased over Australia, northern South America, and Southeast
Asia, while it decreased over the oceans. Increased precipitation in Australia is proven
to be the dominant contributor to the global total sea level change in 2011 (Boening et
al., 2012; Fasullo et al., 2013).

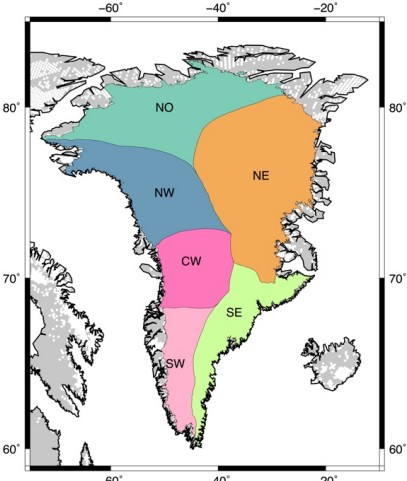

**Figure 1.** Greenland drainage basins. NO: north; NE: northeast; SE: southeast; SW: southwest; CW: central west and NW: northwest according to Rignot Basins from IMBIE 2016 (Rignot et al., 2011). White dots show ice caps in Greenland and surrounding areas.


It is well known that the Greenland ice sheet (GrIS) plays an important role in Earth
system dynamics, which not only affects sea level but also contributes to the elastic
response of the solid Earth. Here, we present detailed mass balance results for the
GrIS drainage basins by estimating the anomalous acceleration of the mass loss and
its contributions to sea level fingerprints (SLF). Figure 1 shows Greenland ice
drainage units, named Rignot Basins from IMBIE 2016 (Ice Sheet Mass Balance
Intercomparison Experiment), which are based on historical usage (Rignot et al.,
2011). The GrIS is divided into six regions based on the glacier regime. Central west



and northwest have a clear basin boundary near Rinks. Central west to southwest
mark the transition from tidewater to land-terminating. Southeast vs northeast chiefly
represents a transition in the surface mass balance (SMB) with a well-defined divide
inland. We use GRACE monthly gravity fields and the monthly cumulative SMB
from the Regional Atmospheric Climate Model (RACMO) to estimate the spatial
distribution of the ice mass balance. The time series of mass changes were estimated
by a mascon fitting method described by Jacob et al. (2012). The relative sea level
(RSL) spatial variations were computed by solving the sea level equation with
self-attraction and loading effects. Based on the above results, we further discuss the
sensitivity kernels and rescaled GrIS time series due to the limitation of exact-defined
basin mask and GRACE resolution; we also analyze spatial variations of the abnormal
melting in glaciers, near-surface air temperature over Greenland and contributions of
GrIS to sea level changes.

**2   Data and methods**
**2.1 GRACE**
The GRACE mission design makes it particularly useful for surface mass variations
studies. GRACE was jointly launched by NASA and the German Aerospace Center
(DLR) in March 2002 (Tapley et al. 2004). The Level-2 gravity products provide
complete sets of spherical harmonic (Stokes) coefficients, typically up to the
maximum degree/order $l_{max}$=120, averaged over monthly intervals. Detection of mass
change using GRACE data becomes a widely used tool for estimation of the ice sheet
mass balance due to the operational difficulties of other measurements over large
areas. However, interpretation of GRACE data is complicated by the intrinsic mixing
of gravity signals. Glacial isostatic adjustment (GIA) can be corrected by modeling
the lithospheric response to loading changes (Velicogna and Wahr, 2006) while other
mass change contributions (e.g., terrestrial water storage) are smaller on ice sheets
compared to other areas.



In this study, we use monthly sets of spherical harmonics from the GRACE Release
05 (RL05) gravity field solutions generated by the Center for Space Research (CSR)
at the University of Texas, spanning January 2003 to December 2015. Each monthly
GRACE field consists of a set of Stokes coefficients, $C_{lm}$ and $S_{lm}$, up to degree and
order ($l$ and $m$) of 60. We replaced the GRACE $C_{20}$ coefficients with the results
inferred from satellite laser ranging (Cheng et al. 2013), and include degree-one
coefficients as calculated by Swenson et al. (2008). The Stokes coefficients from A et
al. (2013) are used to remove the GIA effect.
**2.2  SMB**
In several studies RACMO and the Firn Densification Model (FDM) have been
applied for Greenland using different models at different resolutions and with various
forcing at the boundaries. To further compare and validate the GRACE-derived mass
changes, we use monthly SMB fields to simulate GrIS mass balance from RACMO
version 2.3 (RACMO2.3), which are provided on a grid of about 40 vertical layers
and a horizontal resolution of ~11×11 km$^2$ for the period January 1958–December
2015 (Noël et al., 2015). Then we analyze the spatial and temporal patterns of glacial
dynamics components combining GRACE and SMB data.

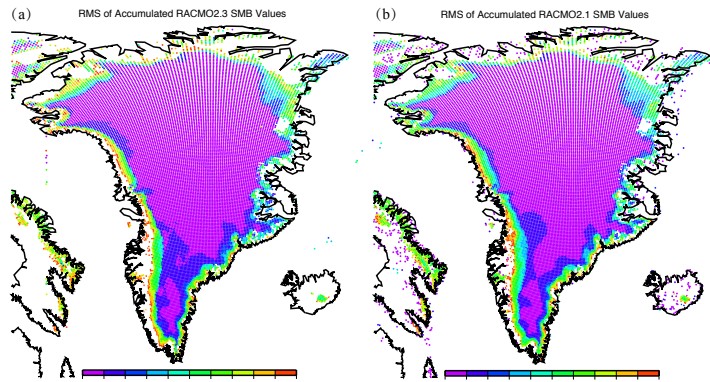

**Figure 2.** Root mean square errors of accumulated SMB values in RACMO2.3 (a)

and RACMO2.1 (b) during 1960 to 2011.



The latest version of RACMO2.3 has been specifically developed to simulate SMB of
glaciated regions as an updated version of RACMO2.1 (Ettema et al., 2009; Van
Angelen et al., 2014). Figure 2 shows root mean square errors of accumulated SMB
values in two versions for the period 1960 to 2011. Both models consist of 312
(latitude) × 306 (longitude) grid cells and include Iceland, the Svalbard archipelago
and the Canadian Arctic. Overall, there is no significant difference in the cumulative
root mean square (1960–2011) between the two versions of the model, but
RACMO2.3 shows larger fluctuations at the boundary of GrIS. This is mainly due to
the fact that RACMO2.3 is forced at the lateral boundaries by the 40-year European
Centre for Medium-Range Weather Forecasts (ECMWF) Reanalysis (ERA-40) for the
period January 1958–December 1979 and the ECMWF Interim Reanalysis
(ERA-Interim) afterwards (van den Broeke et al., 2016).
In this study, we first used the GrIS mask as prescribed in RACMO2.3 to remove
effects of the ice caps from entire SMB in Greenland and integrated them over time to
get accumulated SMB values. Because SMB represents the sum of mass fluxes inside
and away from ice sheets, the mass balance of the grounded ice sheet is governed by
the difference between SMB and the solid ice discharge across the grounding line.
Thus, the ice discharge must be subtracted from the accumulated SMB (SMB minus
ice discharge) to be compared with GRACE (van den Broeke et al., 2016). After
removing the temporal average of the accumulation rates at each point, we convert
SMB data to the spectral domain and truncate them to degree 60, i.e., the limit of the
GRACE data.
**2.3 Other datasets**
Initially, we employed the Noah land hydrology model (version 2) in the Global Land
Data Assimilation System (GLDAS-2) to remove continental water mass
contributions, but we found that there is a large error in the results. The global
GLDAS/Noah, which possesses monthly intervals with a spatial resolution of 1.0
degree, provides a total amount of the water stored in all layers, snow, and canopy, but



does not include the groundwater and water storage changes in rivers or lakes (Rodell
et al., 2004). It also excludes the water storage estimates from the GrIS and
permafrost areas (Liu et al., 2016). Likely, the abnormally large snow values obtained
for Greenland are a result of unreliable forcing data. We simulated mass changes from
the soil moisture component and found that the soil moisture from GLDAS is
dominated by the annual cycle and the annual amplitudes are much smaller than the
GrIS change. Finally, we ignored the terrestrial water storage (e.g., mainly presented
as seasonal changes, no obvious long-term trend) impacts on the mass change in
Greenland and assumed that the mass balance revealed by GRACE data is mainly due
to ice sheet changes.
A previous study based on satellite-derived ice-surface temperature has confirmed a
positive trend of the near surface temperature of GrIS and two major melt events from
2000 to present (Hall et al., 2013). Therefore, we chose the temperature data from the
GLDAS/Noah model, which integrates the latest NASA remote sensing products (e.g.,
moderate-resolution imaging spectro-radiometer, MODIS). We investigated whether
there was clear correlation in spatial distribution of the GLDAS/Noah forcing data
(i.e., temperature) and the GrIS variations spanning from 2003 to 2015.
On climate timescales, the global mean sea level rise is mainly caused by increasing
volume of the global ocean in consequences of thermal expansion) and increasing
ocean mass due to water masses from land (i.e., GRACE-derived barystatic sea level
rise caused by loss of ice and reduction of liquid water storage on land). Reliable time
series of global mean sea level based on satellite altimetry (TOPEX/Poseidon, Jason-1
and OSTM/Jason-2) are available since September 1992 (the global mean sea level
data      was      downloaded      from      NASA,      available      at:
https://sealevel.nasa.gov/understanding-sea-level/key-indicators/global-mean-sea-leve
l). All biases and cross-calibrations have been applied to the data, therefore sea
surface height anomalies derived from various altimetry missions are expected to be
consistent. The data have been presented as changes relative to January 1, 1993



averaged over 2-months intervals. The GIA correction has been applied to the data
(Beckley et al., 2017). To estimate steric sea level anomalies, we used time series of
3-month total steric sea level anomaly data, which is a contribution of the changes in
the global ocean heat storage for the 0–700 m and 0–2000 m layers (the total steric
sea   level   anomaly   data   was   downloaded   from   NOAA,   available   at:
https://www.nodc.noaa.gov/OC5/3M_HEAT_CONTENT/basin_fsl_data.html).
**2.4  Spatial Averaging and scaling factor methods**
Observations of mass variability are, in particular, useful for estimates of changes of
continental water storage. These water storage changes are generally addressed by
constructing specific averaging functions optimized for each region (Swenson and
Wahr, 2002). Note that the averaging kernel method implies a Gaussian averaging
function at each point, and sums those averaging functions expressed as the finite
number of harmonic degrees in the GRACE solution (e.g. $l_{max}$ = 60 for CSR solutions).
Thus, the optimal averaging kernel technique provides an estimate of the total mass
change of the region but does not give accurate estimates of sub-regions, such as
those in Figure 1, due to the spatial resolution of the GRACE data. Therefore, the
effect of mass changes is spread up to several hundred kilometers outside the region.

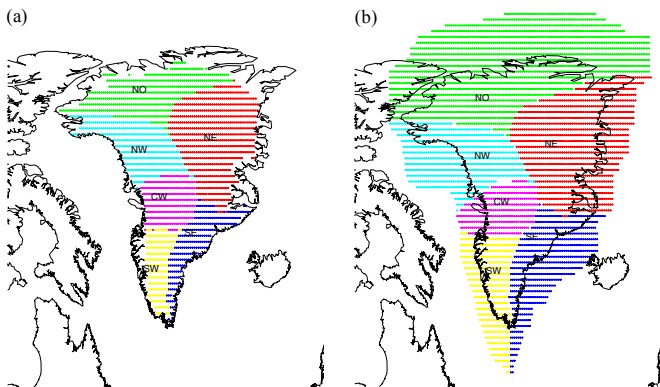

**Figure 3.** Mascons for the GrIS drainage basins (a). Each colored region represents

a single mascon. (b) similar to Figure 3a but for the extended mask of six mascons.



In this case, we applied an approximation mascon fitting method to GRACE and
SMB data to perform a comparison at the regional level. This fitting method is based
on the least squares mascon approach to calculate the averaged time series for each
region (Jacob et al., 2012; Sutterley et al., 2014). To evaluate the spatial differences in
the melting of GrIS at a regional scale, we divided the ice sheet into six extended
mascons as shown in Figure 3, and each mascon was composed of small blocks
defined on a 0.5-degree grid; a unit mass equal to 1 cm of water was distributed
uniformly over the block (Farrell 1972). We applied a 150-km Gaussian smoothing
function on the Stokes coefficients for the GRACE (GIA corrected), SMB and all
mascon coefficients.
We simultaneously fit the extended mascon Stokes coefficients, in which GrIS is
represented by a single basin, to monthly GRACE coefficients (after post-processing
described in section 2.1) to obtain estimates of monthly mass variability for each
mascon. The corresponding result in terms of time series of entire GrIS is shown in
Figure 4. When using extended mascons, the mass loss is assumed to be uniformly
distributed over mascons, which is not the case everywhere (e.g., because there is no
or relatively small mass change over the oceans). Thus, it is necessary to identify a
realistic scaling factor. Assuming that there is a 1 cm uniform layer over exact and
extended GrIS, the total mass is 17.495 Gt and 39.303 Gt, respectively. We used the
exact Greenland mascon as the input to fit the extended mascon to the input signal. In
this way, the 0.537 cm uniform mass is obtained over the extended GrIS, which is
equivalent to a 46% reduction in ice thickness of the input mass, which is in good
agreement with previous studies based on averaging functions extended outside
Greenland (Velicogna and Wahr, 2006). The final scaling factor of the mass inferred is
(39.303/17.495) ×0.537=1.206. Therefore, the mass changes estimated with the
extended mascon are larger by a factor of 1.206 when degree and order of Stokes
coefficients are limited to 60 (Figure 4).



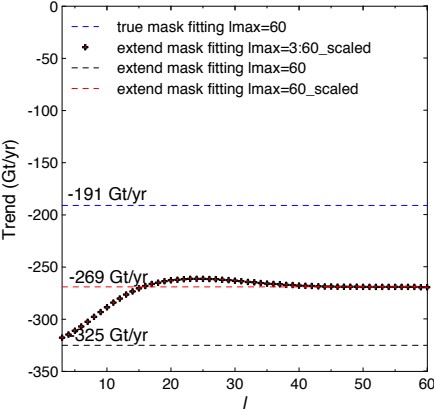

**Figure 4.** Time series for the entire GrIS from the exact and extended mascons to fit monthly GRACE coefficients. Red crosses are scaled extended mascon fitting results due to change of the scale factor for different degree ($l$).


We take into account the fact that the effect of each mascon could smear into the
neighboring ones. Supposing that the mass spread is uniform over the truly $mascon_i$,
we computed the Stokes coefficients from the input mass, and then fit extended
$mascon_k$ to the set of Stokes coefficients. Basing on the scaling method described
above, those values can be used to construct a ratio matrix $A(k,j)$, which is the
contribution of those Stokes coefficients to the result for $mascon_k$. Time series for
selected regions were calculated using the corresponding mascons to fit GRACE
Stokes coefficients. If $M(j)$ are the true mascon values, and $N(k)$ are the values that we
get from the mascon fitting, then the linear observation equations is $N(k) =$
$\sum_{j=1}^{6} A(k,j) \times M(j)$. Therefore, the true mascon values may be solved in a generalized
inversion by $M(j) = A^{-1}(k,j) \times N$. This method not only estimates the total mass
change but also provides time series for each sub-area after the leakage correction.
However, it is worth noting that the extended mascon increases the weight of the
boundary in the sensitivity kernels and also causes external leakage in the fitting
results, e.g., mass change from the external glaciers, ice caps and eustatic sea level.



The sensitivity kernels and leakage effects are explained in details in Section 4.1.
**2.5  Sea level fingerprint**
The global SLF reflects the redistribution of ocean-land masses driven by climate
change; and these load changes cause the elastic structural response of the crust and
affect the viscosity and strength of the lower mantle of the Earth (Peltier and Andrews,
1976). RSL changes, for instance, caused by GIA span over a time scale of 1 to 10000
years. However, for shorter time scales (1 to 100 years), melting of ice sheets, glaciers
and ice caps directly leads to increase of ocean volume and causes instantaneous
elastic deformation of the solid Earth. RSL is the height of the sea surface relative to
the sea floor, which is defined as the difference between the geoid and the crust. The
RSL solution is often referred as the fingerprint of terrestrial mass changes.
In this study we use scaled monthly (1 degree × 1 degree) mass change grids of GrIS
as input to solve the self-consistent sea level equation (Farrell and Clark, 1976; Milne
et al., 1999) and calculate regional SLF due to self-attraction and loading effects
(Tamisiea, 2010) of mass changes on Greenland. We use the load Love numbers given
by Jentzsch (1997), which were calculated using the 1-D PREM elastic Earth model
(Dziewonski and Anderson, 1981). We also consider the Earth rotation feedback but
neglect changes in the coastline and effects of atmospheric and non-tidal ocean
loading for short-term sea level variations during 2003 to 2015.

**3    Results**
**3.1  Spatial GrIS variability**
The spatial pattern of long-term mass trend, shown in Figure 5, was obtained from the
monthly GRACE mass solutions for Greenland from 2003 to 2009 (a), 2010 to 2012
(b), 2013 to 2015 (c) and 2003 to 2015 (d). A clear negative trend was identified
across the entire ice sheet except in high altitude areas (>2000 m) in the central part.
During 2003–2015, the mass loss increased in northwest, central west, south west and
southeast, especially along the north-west coast and the south-east coast. In the north



and northeast, the mass melted relatively slowly compared to the other four areas. The
ice mass loss increased in 2010–2012 and 2013–2015 relative to 2003–2009.
Especially important is that during 2010–2012 a large mass loss is revealed in the
entire southern and western regions of Greenland (Figure 5b), which reflects a major
melting event that took place in this period. For example, the anomalous warm
summer and declined albedos associated with the north Atlantic oscillation led to
increased temperatures over Greenland in 2010 (Box et al., 2012). Consequently, the
extreme melt event took place over almost the entire surface of the GrIS in 2012
(Nghiem et al. 2012).

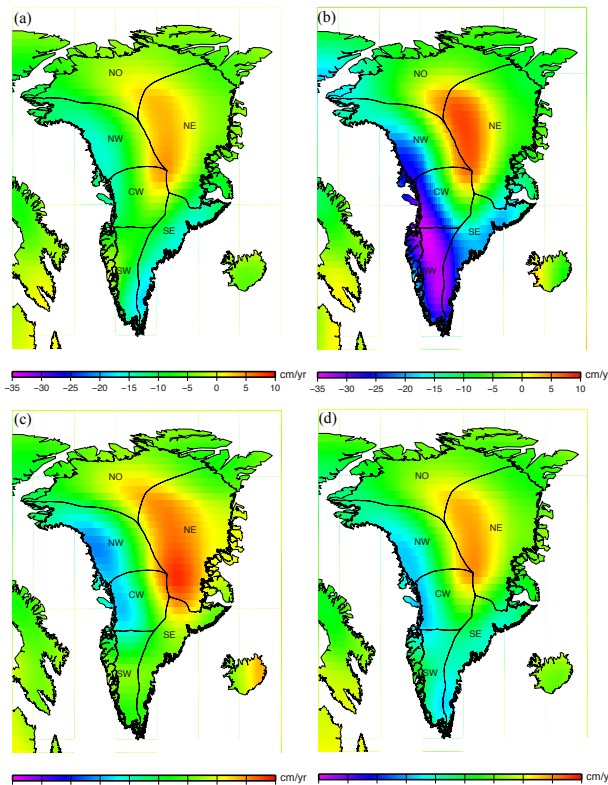

**Figure 5.** GRACE-derived linear trends of GrIS ice mass balance in 2003–2009 (a),

2010–2012 (b), 2013–2015 (c) and 2003–2015 (d).






Figure 6 shows spatial patterns of ice mass changes from SMB data. In 2003–2015,
the SMB results indicate that ice mass loss and thinning was concentrated in the entire
coastline as well as in western and southeast basins of Greenland. In 2010–2012, mass
loss and thinning were stronger in the northwest, central west, south west and
southeast; and this spatial and temporal distribution is very consistent with the
GRACE-derived mass loss. However, the trend magnitude of SMB is smaller than of
the GRACE results. Additionally, we shall keep in mind that the GRACE-derived
results reflect mass changes of both SMB and ice discharge, e.g., beginning at 1995,
SMB decreased while ice discharge increased, due to acceleration of the ice melting
in several large outlet glaciers in the southeast and northwest, which leading to a
quasi-persistent negative mass balance (van den Broeke et al., 2016). Moreover,
because of large runoff and surface mass fluxes (i.e., meltwater and snowfalls) at the
boundary of the GrIS, the current horizontal resolution of RACMO2.3 (11 km) is
insufficient to resolve individual, low-lying outlet glaciers of the GrIS (Noël et al.,
2016), which leads to potentially large errors and uncertainties in accumulated SMB
values (Figure 2).

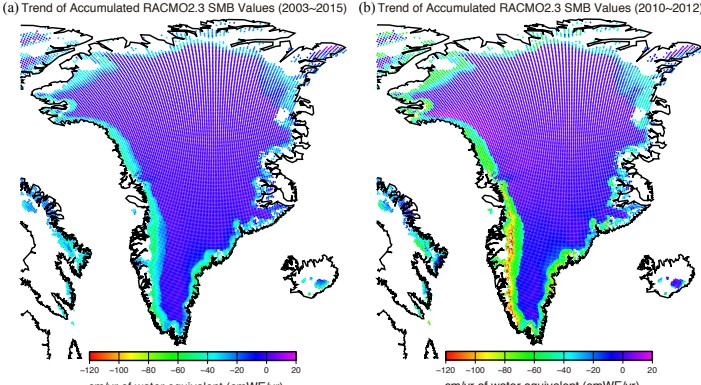

**Figure 6.** SMB trend in millimeter water equivalent per year or (mmWE/yr)
obtained from the RACMO2.3 monthly SMB fields. (a) 2003 to 2015 and (b) 2010
to 2012.



### 3.2 Time series of mass change


In order to obtain time series of GrIS mass changes we applied the basin estimation
and scaling method described in Section 2 (Figure 7). Representing GrIS by single
and extended mascons, we found that the scaled trend rate (–269 Gt/yr when $l_{max}$=60
shown in Figure 4) from 2003 to 2015 in the whole GrIS region is in good agreement
with that reported by –270 Gt/yr during 2003–2012 (Schrama et al., 2014) and –270
Gt/yr during 2003–2014 (van den Broeke et al., 2016). When the GrIS is represented
by six extended basins, the results also show a continuous decrease both before and
after scaling (top and bottom left in Figure 7) from 2003 to 2015; since 2010, the rate
of this decrease suddenly accelerated towards the end of 2012. The rate of the mass
loss obtained by scaled GRACE and SMB is also similar, –288±7 Gt/yr in GRACE
and –275±1 Gt/yr in SMB from 2003 to 2015. The magnitude of the trend increased
significantly over the period 2010–2012, about –456±30 Gt/yr in GRACE and –
464±38 Gt/yr in SMB. The errors here represent fitting uncertainties, while the real
uncertainties are mainly due to the GIA correction, leakage of signal from outside ice
sheet, and GRACE measurement errors. Those effects in the trends were estimated to
be 20 Gt/yr in both time series (Van den Broeke et al., 2009). Our estimates are in
good agreement with the magnitude of the fitted linear trend both from GRACE and
SMB over the period 2003–2014 (van den Broeke et al., 2016) but slightly larger than
the reported GRACE-derived mass loss rate from Sutterley et al. (2014), Velicogna et
al. (2014) and Forsberg (2017). It should be noted that the overestimation of our
results likely comes from the leakage effect of glaciers and ice caps due to the fact
that we used extended mascons to fit the GRACE and SMB data. The impact of this
part may reach about 20~80 Gt/yr (Bolsch et al. 2013; Velicogna et al., 2013).





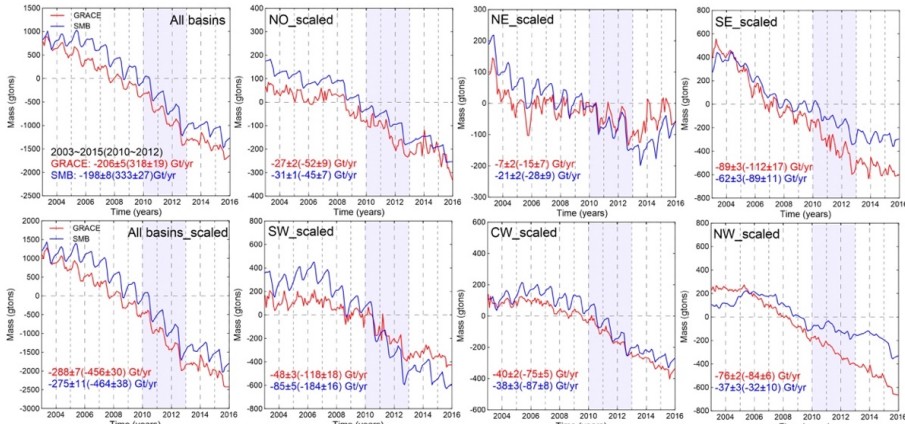

**Figure 7.** Ice mass change in gigatons (gtons) for GrIS, the top part of the figure from left to right is from the exact mascons of GrIS, extended mascons (after scaled) of NO, NE and SE, respectively. The lower part of the figure from left to right is from the extended mascons (scaled) of GrIS, SW, CW and NW, respectively. GRACE time series for January 2003 to December 2015 (red), time series of cumulative SMB anomaly for January 2003 to December 2015 (blue). Light blue bands represent the time span from January 2010 to December 2012.


For GrIS drainage basins at the regional scale, the melting rate of GrIS in the southern
part is significantly higher than in the northern part. The mass loss in the north and
northeast was less than –31 Gt/yr for both GRACE and SMB during 2003–2015, and
the mass loss of the other four basins (i.e., northwest, central west, south west and
southeast) were several times larger than the ones in the two northern regions. The
time series of GRACE and SMB revealed that almost all regions experienced large
mass losses in 2010–2012. In the southwest and southeast, we found an anomalous
acceleration of the mass loss of –118±18 Gt/yr and –112±17 Gt/yr in GRACE and –
184±16 Gt/yr and –89±11 Gt/yr in SMB, respectively. The contribution of these two
regions is responsible for about 50% of the total loss. In addition, we also found that
the melting rate of ice sheets from SMB was greater than the estimates derived from
GRACE in the southwest and northeast. This difference indicates that SMB may




overestimate ice mass changes, since the modeled surface meltwater increases
strongly with decreasing elevation and latitude in the low-lying parts of the
southwestern GrIS (van den Broeke et al., 2016). In addition, the surface ice elevation
was changed by fast-flowing ice dynamics in the southwestern and northeastern areas
(Hurkmans et al., 2014). Since 2013, the mass loss slowed down and recovered in the
GrIS drainage basins. The agreement between GRACE and SMB results also confirm
that the ice sheets returned to near-normal melt conditions, i.e., the refreezing process
reduced the melt extent back to normal conditions (Nghiem et al., 2012).
**3.3  Sea level fingerprints induced by GrIS**
The distribution of GrIS mass changes directly affects the combined contributions of
the sea level self-attraction and loading as well as of the ocean-land mass balance
resulting in differences in the global sea level distribution (Figure 8). Melting of ice
sheets is confirmed over entire Greenland, especially in the southern part and along
the coasts (Figure 6). This mass loss of GrIS caused RSL lowering in the entire Arctic
Circle, for instance, negative changes of RSL in Scandinavia and Northern Europe up
to about –0.6 cm/yr (Figure 8a). It should be noted that the mass loss of Greenland
mainly increases RSL in tropical and southern latitudes due to the isostatic rebound of
the sea floor around Greenland (Figure 8b).

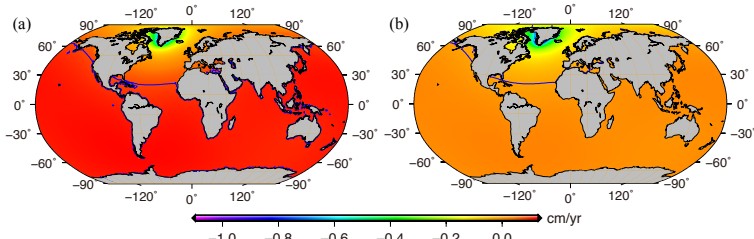

**Figure 8.**  Trends in the sea level fingerprint (SLF) due to mass change of GrIS (a).
(b) contributions from the Earth's elastic response. Trends are calculated for the
time period January 2003 to December 2015. Blue contour in Figures 8a and 8b is
the mean RSL or barystatic sea level equivalent.


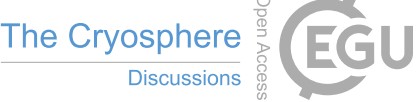



Due to ice sheet melting, the sea level along coastlines located up to 2000 kilometers
away falls as a result of the isostatic uplift of the crust. The escaping seawater flows
across the equator, i.e., the melting of Greenland impacts the coastline of Brazil and
the melting of Antarctica affects the United States. These regional differences are
significant if we consider the global melting of ice sheets, glaciers and ice caps. For
instance, the amount of ice mass melt in the northern hemisphere is higher than in the
southern hemisphere, resulting in apparent RSL rise in the South America, South
Africa, and Australia, what is nearly 30% higher than the global mean sea level rise
rate (Mitrovica et al., 2001; Bamber et al., 2009). In addition, induced by the mass
loss of GrIS, the mean RSL trend is approximately 0.07 cm/yr extending through
Alaska, Mexico and northern Africa (solid blue line in Figure 8). This pattern
illustrates that the dynamic sea level change is determined by the ocean-land mass
redistribution and by the instantaneous elastic response of the lithosphere.

**4    Discussions**
**4.1 Sensitivity kernels and rescaling**
As an example of the averaging kernel, Figure 9 shows the sum of the sensitivity
kernels for all exact and extended mascons shown in Figure 3. Ideally, the solution for
mascon fitting would recover the true spatial average of the mascons' mass. When
mascons are fitted for the exact-defined GrIS drainage sub-areas (Fig. 3a), the results
are automatically scaled. The effective scaling factor based on the least squares
mascon approach is defined assuming that surface masses are spread uniformly across
any mascon. This method will give exactly the right total mass for that mascon, and
will give 0 for the other mascons. However, similar to the optimal averaging kernel
method, the mascon fitting based on an exact-defined basin mask (i.e., truly six
drainage basins) will also cause weakening of the signal or large uncertainty (e.g.,
leakage and bias). This is especially the case in boundary areas, which largely
contribute to the mass loss, because of the finite number of harmonic degrees in the





GRACE solution. Previous studies suggest that an increasing of the number of
mascons covering the anomaly might reduce leakage, so that the anomaly is almost
constant across each individual mascon (Jacob et al., 2012). However, there are also
indications that using more and smaller mascons can lead to the drawback that the
inversion relies more on the higher harmonic degrees.
For six sub-areas of the extended mascon (Figure 9b), we assessed a potential impact
of the non-uniformity over the exact mascons and external mascons. For the leakage
effects, we first computed the mascon distribution between sub-regions, and then we
obtained the scale factors by fitting the six extended mascons to the corresponding
exact mascons (Table 1). To confirm the validity of signal recovery based on this
scaling method, we also used two different regional average methods to compare the
results. One method represents a data-driven approach, which is able to restore the
GRACE signal loss due to filtering independent of the catchment size (Vishwakarma
et al., 2016; 2017). Another method implies scaled optimal averaging functions to
recover unbiased mass estimates for six basins (Velicogna and Wahr, 2006).

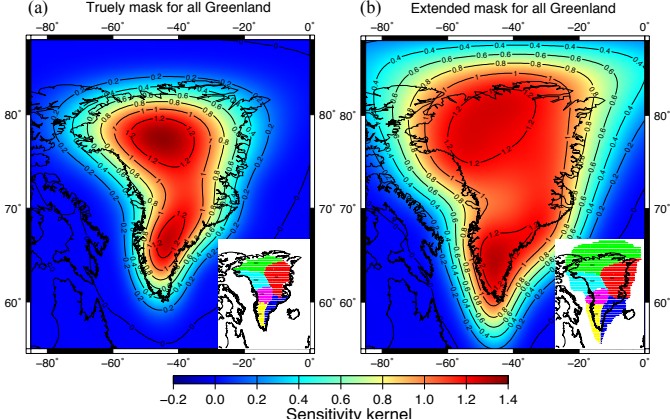

**Figure 9.** Sensitivity kernel for the truly mask (a) and extended mask (b) of all
drainage basins.






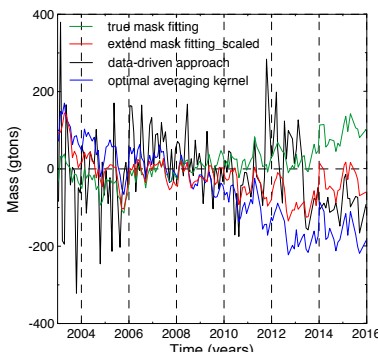

**Figure 10.** Time series of the true mask fitting mascon and scaled extended mascon fitting in the northeast accompanied by the regional average based on the optimal averaging kernel and data-driven approach.


Exemplarily, Figure 10 shows a validation with the time series comparison between
the results from the exact mascon fitting and the extended mascon fitting after
rescaling in the northeast. The results confirm that the exact mascon fitting cannot
accurately extract the melting contribution of glaciers close to the border (i.e.,
sensitivity kernel less than 1 shown in Figure 9a). Consequently, the time series from
the exact mascon fitting in the northeast show an increasing trend, what is
inconsistent with the actual situation and contradicts most previous studies (Velicogna
et al., 2014; Sutterley et al., 2014). In addition, the time series obtained by the other
two methods also confirm the mass loss trend of ice sheets in the northeast. However,
the optimal averaging kernel after scaling may include leakage in other regions and a
data-driven approach shows a large noise error in the time series. This is mainly due
to the fact that the optimal averaging kernels were created to isolate the gravity signal
of individual regions while simultaneously minimizing the effects of GRACE
observational errors and contamination from dynamic changes of nearby glaciers
(Swenson and Wahr, 2002). Though, this method cannot prevent leakage from
adjacent areas. Therefore, there still exists large signal loss in each region due to the
filtering and truncation of GRACE coefficients. A data-driven approach was



developed to extract leakage information from the filtered versions of the field, but
this method also suffers several limitations, e.g., it does not work with sufficient
accuracy for active catchments, and both the scaling factors and the aggregated noise
over catchments increase as the catchment size decreases (Vishwakarma et al., 2016).

**Table 1.** Scale factors of six basins derived with the extended fitting approach

|  | NO | NE | SE | SW | CW | NW |
|---|---|---|---|---|---|---|
| NO_extended | 0.952 | 0.014 | 0.000 | 0.011 | −0.005 | 0.062 |
| NE_extended | 0.126 | 1.063 | 0.059 | −0.031 | 0.056 | 0.112 |
| SE_extended | −0.007 | −0.021 | 0.954 | 0.190 | 0.071 | −0.013 |
| SW_extended | 0.012 | −0.003 | 0.071 | 0.960 | −0.098 | −0.012 |
| CW_extended | −0.042 | 0.036 | 0.151 | 0.136 | 1.045 | 0.050 |
| NW_extended | 0.181 | 0.049 | −0.039 | −0.033 | −0.008 | 0.964 |
| Ratio of total mass to input mass | 1.223 | 1.138 | 1.196 | 1.235 | 1.061 | 1.163 |


## 4.2  Spatial differences of abnormal melting in glacier dynamics

If we ignore the GIA correction error, total mass changes detected by GRACE contain
a component caused by changes in SMB (corrected ice discharge) and a component
caused by ice dynamics. Usually, the latter can be estimated from satellite altimetry
data. Thus, the residuals obtained from GRACE after removing SMB may well reflect
glacial dynamics. Figure 11 shows the residuals for each drainage basin and the entire
GrIS, which is used to interpret the contribution from glacial dynamics to total ice
mass changes. The time series for six drainage basins are quite different and show no
overall trend characteristics in GrIS. In the southeast and northwest, there is a
negative trend in the difference GRACE minus SMB. Global navigational satellite
system data also revealed intense Greenland melting. For example, crustal motion
data show that solitary seasonal waves are associated with substantial mass transport
through the Rink Glacier in 2010 and 2012 (Adhikari et al., 2017). In contrast, a





positive rate of mass change is found in southwest and northeast areas. In central west,
north and entire Greenland, the time series of the residuals do not have apparent
trends. This spatial difference is in a good agreement with surface elevation changes
derived from ICESat, GRACE and GPS data based on previous results (Howat et al.,
2008; Khan et al., 2010; Hurkmans et al., 2014). Particularly, satellite observations
such as the Oceansat-2 satellite, MODIS and Special Sensor Microwave
Imager/Sounder reveal that melt occurred at or near the surface of GrIS across 98.6%
of its surface on 12 July 2012 (Nghiem et al., 2012).

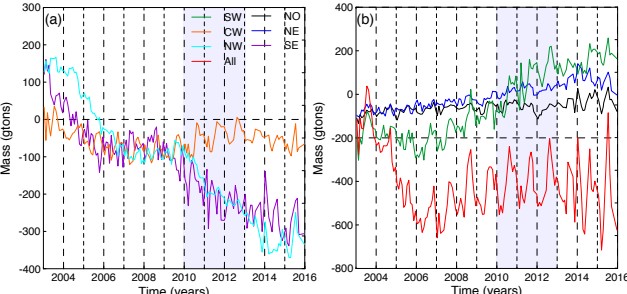

**Figure 11.** Residuals obtained from GRACE after removing SMB for each drainage basin and the entire GrIS.


Because of the combination of the modelled (SMB) and observed (GRACE) data, any
uncertainty or error of the data source will appear in the residuals. Based on the mass
budget method, the SMB model estimates the difference between individual mass
sources (mainly snowfall) and sinks (mainly meltwater runoff and solid ice discharge)
(van den Broeke et al., 2016). The accumulation/ablation zones of an ice sheet are
largely driven by changes in weather conditions (Hanna et al., 2011). More
importantly, glacial dynamics refer to the flow of ice from the interior of the ice sheet
outward through outlet and land-terminating glaciers (Liu et al., 2016). Although this
kind of ice discharge may not be accurately estimated by the SMB model, its
contribution to the total mass balance cannot be ignored either. Another factor





influencing the residual is the accuracy and limited resolution of GRACE data, e.g.,
measurement errors, GIA correction, leakage effects from outside the ice sheet and the
eustatic sea level, etc. For Greenland the uncertainties in the GRACE estimates of the
ice sheet mass balance have been analyzed in previous studies (Van den Broeke et al.,
2009; Bolsch et al. 2013; Velicogna and Wahr, 2013). Therefore, we will not discuss
them here in detail. At the same time, we are aware that the errors come mostly from
the uncertainty in the scaling factor due to partitioning of GrIS into six mascons. The
difference between the non-uniform distribution of actual ice sheets and our
assumption of uniform mass distribution within the basin or each mascon also leads to
uncertainty of the scaling factor, which increases the uncertainty of final mass loss
estimates.
**4.3 Near-surface air temperature over the Greenland**
In general, mass changes of the GrIS mainly depend on temperature variations, which
cause both ice discharge and surface meltwater runoff. Near-surface temperatures can
be derived from global land surface models forced with atmospheric data (e.g.,
Satellite-derived MODIS data in this study) (Syed et al. 2008). Figure 12 shows the
averaged near-surface air temperatures from the GLDAS forcing (i.e, MODIS) data in
Greenland for the periods 2003−2015 (Figure 12a) and 2010−2012 after removing
the average of 2003−2015 (Figure 12b). The spatial distribution of the temperature
anomalies indicates that the increased mass loss rate from GRACE observations and
SMB simulations is mainly due to relatively high surface temperature of South
Greenland (i.e., mean change range from about −10 to −5 ℃, Figure 5d and Figure
6a). According to Figure 12b, there are large positive temperature anomalies over
most parts of Greenland during 2010−2012, which is consistent with the acceleration
of mass loss in the GrIS during the same period.

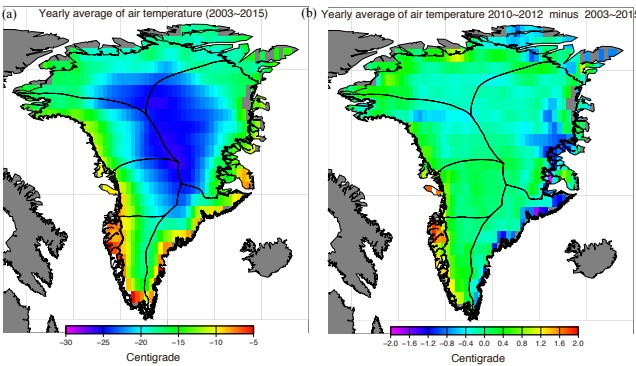

**Figure 12.** Average near-surface air temperatures from MODIS data in Greenland for the periods 2003−2015 (a) and 2010−2012 after removing the average of 2003 −2015 (b).


In response to positive near-surface temperature anomalies in the years 2010 and 2012,
the GRACE and SMB results show accelerated mass loss (Figure 7). In previous
studies, Nghiem et al. (2012) and Hall et al. (2013) already described the major melt
event in 2012 in details, which was captured by ice melt maps from three different
satellite missions. Seasonal and interannual variations in GRACE time series are
qualitatively well reproduced including the large summer mass losses in 2010 and
2012 (van den Broeke et al., 2016). In fact, near-surface air temperatures are most
appropriate for making long-range predictions of ice melting caused by climate
variability. Differences in mass loss between GRACE and SMB are partly attributed
to differences in the temperature input of the SMB model. Although not demonstrated
in this study, the use of corrected SMB inputs based on in situ data will provide more
accurate results when SMB outputs (i.e., sum of mass fluxes towards and away from
the surface ice sheets) are used to refine the vertical and horizontal resolutions of
GRACE. In turn, this reduces the uncertainty in the GRACE-based estimates of mass
changes from ice sheets.
**4.4  Contribution of GrIS to sea level change**
It is well-known that global mean sea level variations are dominated by thermal





expansion caused by heating of the global ocean, and variations of total ocean mass
due to varying water mass fluxes from land to oceans. Here, we attempt to find the
contribution of the GrIS to present-day global mean sea level rise. As shown in Figure
13, the sum of ocean mass variations from GRACE-derived total land contributions
and steric sea level from the total steric sea level anomaly data are close to the
observed sea level trend of 3.3 mm/yr derived from sea surface height anomaly data.
The trend rate of the contributions of the total land (without Greenland), GrIS and
steric sea level changes are 1.1 mm/yr, 0.7 mm/yr and 1.4 mm/yr, respectively.

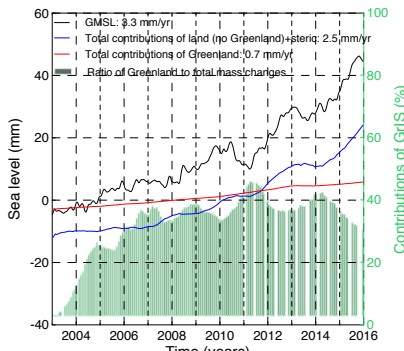

**Figure 13.** Global mean sea level (GMSL) from altimetry during 2003-2015 (black line), total freshwater input from land (without Greenland) and steric sea level changes (blue line), and GrIS contribution (red line). Seasonal signals have been removed. The grey vertical bars show the contribution rate of GrIS to the total mass change (when GRACE data are available).

It is important to note that a V-shaped or solitary wave sea level change is observed
from 2010 to 2012 (black line in Figure 13), which is mainly caused by terrestrial
water storage anomalies (blue line in Figure 13) related to the 2010/2011 La Niña
event (Boening et al., 2012; Fasullo et al., 2013). The GrIS is an important contributor
to present-day global mean sea level rise. The average contribution rate (ratio of GrIS
to the total mass change) is about 31%. Furthermore, there is a clear acceleration of





the proportion of melting in Greenland (grey vertical bars). It might be stressed that
the contribution of GrIS experienced an opposite V-shaped change during 2010-2012,
i.e., the sea level changes from rising to falling. This result indicates that increased
melting of GrIS partially compensated the sea level drop, which was due to a
temporary shift of water from the ocean to continents.

**5    Conclusions**
In this study, the GrIS variations estimated from GRACE gravity fields and SMB data
have been investigated with respect to ice melting of Greenland and its contributions
to sea level changes. The spatial pattern of both long-term mass trends obtained from
monthly GRACE data and SMB indicates that the ice loss appears clearly over
drainage basins in different spatial scales and different time spans. Specifically during
the warm period 2010 to 2012, an anomalous acceleration of mass loss occurs in the
entire southern and western regions of GrIS, which reflects the major melt event due
to higher near-surface temperatures. We calculated time series for six sub-regions
defined by mascons using the least squares mascon fitting approach.
We found that the GrIS changes from the extended mascons solutions combined with
the matrix scaling factor method are in good agreement with previous studies. The
rate of the mass loss obtained by scaled GRACE and SMB is $-288\pm7$ Gt/yr and $-$
$275\pm1$ Gt/yr, respectively, from 2003 to 2015. The magnitude of this trend increased
significantly to $-456\pm30$ Gt/yr in GRACE and $-464\pm38$ Gt/yr in SMB in the period
2010–2012. The residuals obtained from GRACE after removing SMB may reflect
the contribution from glacial dynamics to total ice mass changes. These spatial
differences in the residuals among six drainage basins are in good agreement with the
surface elevation change rates previously derived from the ICESat data.
We computed SLF due to the ice mass fluxes of Greenland for the time period 2003–
2015. RSL anomalies caused by dynamics of the GrIS are not uniformly distributed
across the global oceans due to self-attraction and loading effects. Mass loss of the



GrIS induces reduction of RSL at most coasts of Scandinavia and Northern Europe
(up to about −0.6 cm/yr), In contrast, RSL rise is concentrated around South
America. The contribution ratio of GrIS to total sea level rise increased and the
average contribution rate was about 31% from 2003 to 2015. Although the
contribution of GrIS has an opposite V-shaped change relative to the sea level changes
during 2010−2012, it could not compensate completely the mass transfer from
oceans to the continents.
We also assessed a potential impact of the spherical harmonic truncation, spatial
averaging of mascon fitting and leakages from other time-dependent signals. The
sensitivity kernels for all extended mascons indicate that the sum of kernels is
well-localized to their regions and increased the weight of the boundary of GrIS. This
study suggests that the rescaled GrIS time series based on a uniform distribution
within the basin can effectively reduce the uncertainty caused by non-uniform mass
distribution of continental and oceanic areas. However, contributions of leakage
effects from outside ice sheets and the eustatic sea level to the total mass errors cannot
be avoided when using extended mascons. These factors likely limit the accuracy of
the estimated GrIS contributions to sea level changes.

*Code and Data availability.* The GRACE solutions used in this study are available
from    CSR    (ftp://podaac.jpl.nasa.gov/allData/grace/L2/CSR/RL05/)    and    the
GLDAS/Noah model data is provided by the NASA Goddard Earth Sciences Data and
Information Services Center (http://disc.sci.gsfc.nasa.gov/). Prof. Michiel R. van den
Broeke for providing RACMO v2.1 and v2.3 SMB fields over Greenland produced by
the      Institute      for      Marine      and      Atmospheric      Research
(https://www.projects.science.uu.nl/iceclimate/models/). Vishwakarma et al. (2017)
for providing the MATLAB implementation of the data-driven approach at:
http://www.gis.uni-stuttgart.de/research/projects/DataDrivenCorrection/. The global
mean      sea      level      data      was      downloaded      from      NASA



(https://sealevel.nasa.gov/understanding-sea-level/key-indicators/global-mean-sea-lev
el). The total steric sea level anomaly data was downloaded from NOAA
(https://www.nodc.noaa.gov/OC5/3M_HEAT_CONTENT/basin_fsl_data.html).    We
would encourage interested persons to contact the authorship, who are open to
providing advice and sharing data and code where possible.

*Author Contributions*. Linsong Wang conceived the original experiments, performed
the main data processing and analysis; Liangjing Zhang helped with the data
processing and improve the experiment; Linsong Wang wrote the manuscript;
Liangjing Zhang, Chao Chen, Maik Thomas and Mikhail Kaban contributed to the
discussion of the results and revising the manuscript.

*Competing interests*. The authors declare that they have no conflict of interest.

*Acknowledgments*. We are grateful to Prof. Michiel R. van den Broeke for providing
SMB model data. We also thank Dr. Vishwakarma for giving us more help to use the
code. This work is supported by the National Natural Science Foundation of China
(41504065, 41574070 and 41604060) and the Fundamental Research Funds for the
Central Universities, China University of Geosciences (Wuhan).

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
