# Peer review of "Anomalous acceleration of mass loss in the Greenland ice sheet"

_The Cryosphere, 2018_

## Referee Comment (RC1) · Anonymous Referee #1 · 11 Oct 2018

Review of paper "Anomalous acceleration of mass loss in the Greenland ice sheet drainage basins and its contribution to the sea level fingerprints during 2010-2012" by L. Wang et al., submitted to The Cryosphere Discussions.

The authors make use of 13 years of GRACE data to infer mass changes over the Greenland ice sheet, while focusing on a period of exceptional mass loss (2010-2012). By means of a mascon approach they study the individual contribution of six drainage basins, as well as the impact of the total mass loss on global and regional sea level rise. The paper is nicely written and clearly organized.

However, I think that both methodology and main findings add very little to what is already known in the literature. As such, I unfortunately don't think that this manuscript deserves publication in The Cryosphere.

Below, I am concentrating on a few major objections.

First, the methodology. The authors follow the approach by Jacob et al. (2012), which is rather outdated. Since then, several more advanced mascon approaches have been published (e.g., LuthckeÂăetÂăal.2013; SchramaÂăetÂăal.2014; Watkin-sÂăetÂăal.2015; Ran et al., 2018), and some of the resulting solutions are even publicly distributed (e.g., by NASA JPL and NASA GSFC). In particular, I am rather sceptical about the use of relatively large drainage basins, each of them including both interior and peripheral regions (where the physical processes driving mass changes are very different), especially because the method is based on the determination of scaling factors inferred from uniform test mass distributions. The authors are honest about the limitations of their approach (lines 478-483). Still, those limitations largely undermine the impact of the presented results, and their value is not supported by the comparison with recent literature (lines 327-332).

Secondly, the role of the 2010-2012 mass change seems overrated. Even though Figure 5 clearly shows an increased loss in the SW region, the timeseries of Figure 7 also show that the rates of the total Greenland contribution were not particularly different from the previous years. A regional study would still be useful, for example in combination with more advanced modelling work.

Thirdly, the study of the sea level impacts. The computation of sea level fingerprints due to Greenland has been published several times before, starting from the work of Mitrovica et al. (2001). So, sections 2.5 and 3.3 are really not presenting anything new. The discussion of the impact on global mean sea level in Section 4.4 is somehow interesting, but also there I do not share the conclusions of the authors about the 2010-2012 period. After looking at Figure 13, it seems to me that the rather constant

TCD
Greenland contribution (red line) is simply having a larger weight on the global values because of the 2011 sea level low: that's not the same as saying that Greenland had any mitigating effect.

Finally, it is not very clear to me what is this study adding to the results already published by Velicogna et al. (2014), that includes similar figures about the regional mass change, as well as timeseries up to the end of year 2013.

---

## Referee Comment (RC2) · Anonymous Referee #2 · 5 Nov 2018

The title of this paper is "Anomalous acceleration of mass loss in the Greenland ice sheet drainage basins and its contribution to the sea level fingerprints during 2010-2012." From this title, I was intrigued to see if the authors found evidence of unique sea level fingerprints that arose in 2010-2012 due to specific melting events from Greenland basins.

I was sorely disappointed. The majority of the paper is instead a rehash of how to compute area mascons over Greenland from satellite gravity data, following previously documented procedures. There is absolutely nothing new here. The authors then take

these mascons, add them together to compute the total mass loss over Greenland between 2003 and 2015, and compare to global mean sea level and fingerprints. Again, there is absolutely nothing new here and the work has been presented much better in numerous other papers.

The discussion of "acceleration" in 2010-2012 (Sections 3.1 and 3.2, Figures 5, 6, and 7) is superficial, not quantitative, and has been already discussed in other papers (i.e., vellicogna et al., 2014). There is absolutely no discussion of fingerprints from this anomalous period of 2010-2012.

The presentation of the methods was also poor. As someone who understands all these procedures, I was confused in many places. I would not want a novice to try to understand the methods based on this paper. They should instead go to the original (and better written) papers.

Overall, there is nothing in the manuscript that would make it worthy of publication in "The Cryosphere."